# Fibrosis Severity in MASLD Determines the Predictive Value of Lp-PLA2 for Carotid Atherosclerosis in Type 2 Diabetes: A Cross-Sectional Study

**DOI:** 10.3390/biomedicines13102431

**Published:** 2025-10-05

**Authors:** Junzhao Ye, Rui Song, Xiaorong Gong, Xin Li, Congxiang Shao, Bihui Zhong

**Affiliations:** 1Department of Gastroenterology, The First Affiliated Hospital, Sun Yat-sen University, No. 58 Zhongshan II Road, Yuexiu District, Guangzhou 510060, China; 2Department of Gastroenterology, First Affiliated Hospital, Guangzhou Medical College, Guangzhou 510060, China; 3Department of Gastroenterology, Affiliated Dongguan People’s Hospital, Southern Medical University (Dongguan People’s Hospital), Dongguan 510515, China

**Keywords:** metabolic-dysfunction-associated steatotic liver disease, lipoprotein-associated phospholipase A2, carotid atherosclerosis, type 2 diabetes mellitus

## Abstract

**Background:** Elevated Lp-PLA2 activity, a marker of inflammation and oxidative stress, is linked to increased cardiovascular disease (CVD) risk in type 2 diabetes mellitus (T2DM). Given that high Lp-PLA2 activity is a hallmark of metabolic-dysfunction-associated steatotic liver disease (MASLD), we aimed to investigate whether it contributes additional CVD risks when MASLD coexists with T2DM. **Methods:** This study included 1095 patients with T2DM, consecutively enrolled at the First Affiliated Hospital, Sun Yat-sen University, between June 2020 and November 2022. Liver steatosis and stiffness were assessed via abdominal ultrasound/CT and fibrosis-4 (FIB-4) scores, respectively. Carotid atherosclerosis (CAS) was defined as the presence of intima-media thickening or carotid plaque and was evaluated using high-resolution B-mode ultrasonography. **Results:** Among 674 MASLD patients, higher levels of Lp-PLA2 activity were observed compared to those in 421 non-MASLD individuals (573 ± 164 U/L vs. 540 ± 170 U/L, *p* = 0.002), while no association was found between steatosis degree and Lp-PLA2. Lp-PLA2 levels exceeding a threshold of 570 U/L were identified as a risk factor for CAS, with each one standard deviation increase in Lp-PLA2 corresponding to an odds ratio of 2.67 (95% confidence interval: 1.31–5.42, *p* = 0.007), while a similar association was not observed in patients with normal FIB-4 levels. **Conclusions:** Elevated Lp-PLA2 activity is associated with MASLD and insulin resistance in T2DM, while Lp-PLA2 was not related to the degree of liver steatosis. A threshold of 570 U/L is associated with CAS risk, specifically in those with concurrent advanced liver fibrosis, highlighting the potential role of Lp-PLA2 in cardiovascular risk stratification in this subset but within the limitations of a cross-sectional study.

## 1. Introduction

Metabolic-dysfunction-associated steatotic liver disease (MASLD) encompasses a spectrum of liver conditions induced by metabolic stress, ranging from simple steatosis to steatohepatitis, cirrhosis, and hepatocellular carcinoma. Additionally, MASLD is a significant contributor to extrahepatic metabolic complications, including diabetes and cardiovascular disease (CVD) [1]. The coexistence of MASLD and T2DM is highly prevalent and creates a synergistic adverse effect, accelerating the progression of both liver and cardiovascular damage. This interplay is driven by shared pathophysiological mechanisms, including insulin resistance, chronic inflammation, and dyslipidemia, which collectively promote the development of atherosclerosis—a key feature of CVD [2].

Atherosclerosis, particularly in the carotid arteries (carotid atherosclerosis, CAS), is a major cause of morbidity and mortality in patients with T2DM. The disease is marked by abnormalities in serum lipids and vascular inflammation [2]. Lipoprotein-associated phospholipase A2 (Lp-PLA2) serves as a vascular-specific inflammatory marker that plays a significant role in the progression of atherosclerosis [3]. As an enzyme that catalyzes the hydrolysis of oxidized phospholipids on the surface of low-density lipoprotein cholesterol (LDL-c) particles, Lp-PLA2 produces pro-inflammatory mediators, such as lysophosphatidylcholine and oxidized nonesterified fatty acids [4], positioning it as an independent predictor of coronary artery disease [5].

The liver, a primary organ responsible for metabolic homeostasis, also plays a crucial role in the clearance of circulating Lp-PLA2 [6,7,8,9]. Importantly, the ability of Lp-PLA2 to predict CVD in individuals with T2DM appears to be influenced by the presence of concomitant metabolic disorders, such as MASLD. However, previous studies have reported conflicting results regarding changes in Lp-PLA2 concentrations in MASLD, and it remains unclear whether Lp-PLA2 activity, which more accurately reflects its biological role, is affected by MASLD status [10,11]. Moreover, whether MASLD, especially its more advanced fibrotic stages, modifies the association between Lp-PLA2 and atherosclerosis in the high-risk T2DM population is unknown.

Therefore, we aimed to investigate the interrelationship between Lp-PLA2 activity, MASLD, and CAS in patients with T2DM. Specifically, the primary aim of this study was to investigate whether Lp-PLA2 activity predicts carotid atherosclerosis in patients with T2DM and MASLD and to determine whether this relationship is modified by the severity of liver fibrosis.

## 2. Methods

### 2.1. Study Design and Population

This study prospectively enrolled adult patients with T2DM who were admitted to the inpatient wards of the First Affiliated Hospital of Sun Yat-sen University between June 2020 and November 2022. All participants were thoroughly informed about the study’s objectives and provided their consent by signing informed consent forms. The study protocol conformed to the ethical guidelines of the 1975 Declaration of Helsinki, complied with TREND guidelines and received approval from the Medical Ethics Committees of the First Affiliated Hospital, Sun Yat-sen University, and was registered with the Chinese Clinical Trial Registry under Ethical Approval Code [2020]187 and Registration Number ChiCTR2000034197 (https://www.chictr.org.cn/searchproj.html) on 28 June 2020. The exclusion criteria included (1) concurrent viral hepatitis, alcohol over-consumption (defined as an average intake of >30 g/day for men and >20 g/day for women over the 3 months prior to study inclusion), autoimmune liver diseases, or other conditions unrelated to MASLD; (2) the presence of malignant tumors; (3) severe kidney dysfunction; (4) admission to the ICU during hospitalization; (5) acute diabetes complications, such as diabetic ketoacidosis or a hyperosmolar hyperglycemic state; and (6) pregnancy [12,13].

### 2.2. Diagnosis Criteria

#### 2.2.1. T2DM

According to the Chinese Type 2 Diabetes Mellitus Guidelines (2020 edition) [12], individuals exhibiting typical symptoms—such as polyuria, polydipsia, polyphagia, and unexplained weight loss—must meet at least one of the following criteria: a random blood glucose level of ≥11.1 mmol/L; a fasting plasma glucose level of ≥7.0 mmol/L after a minimum of 8 h of fasting; a 2 h plasma glucose level of ≥11.1 mmol/L during an oral glucose tolerance test; or a glycated hemoglobin (HbA1c) level of ≥6.5%.

#### 2.2.2. MASLD

Patients were confirmed to meet the 2024 EASL criteria for diagnosis, which require the presence of hepatic steatosis alongside at least one cardiometabolic risk factor [14]. The diagnosis was further supported by observed liver fat changes identified through an ultrasound scan or a CT scan.

#### 2.2.3. MS

According to the Chinese T2DM Prevention and Treatment Guidelines (2020 edition) [12], a diagnosis of MS is established when at least three of the following criteria are met: (1) abdominal obesity, defined as a waist circumference of ≥90 cm for men and ≥85 cm for women; (2) T2DM; (3) hypertension, indicated by a blood pressure of ≥130/85 mmHg and/or a history of treatment for hypertension; (4) fasting triglyceride levels of ≥1.70 mmol/L; and (5) fasting high-density lipoprotein-cholesterol (HDL-C) levels of <1.04 mmol/L.

### 2.3. Clinical Estimation

General medical history information, including age, gender, duration of diabetes and treatment, and history of underlying diseases and medications, as well as smoking and drinking habits, was gathered using structured questionnaires. The duration of diabetes is defined as the time elapsed since the onset of typical symptoms, laboratory tests indicating elevated blood sugar levels, or a prior diagnosis of diabetes. Anthropometric measurements were also collected, encompassing height, weight, blood pressure, and waist circumference. Body mass index (BMI) is calculated as weight (kg) divided by height squared (m^2^). Waist circumference is measured at the midpoint between the axillary midline rib edge and the iliac crest.

### 2.4. Laboratory Determinations

Fasting venous blood was drawn to measure alanine aminotransferase (ALT), aspartate aminotransferase (AST), uric acid (UA), creatinine (Cr), total cholesterol (CHOL), triglyceride (TG), LDL-c, high-density lipoprotein cholesterol (HDL-c), fasting blood glucose (FBG), and glycated hemoglobin (HbA1c) levels and the urinary albumin/creatinine ratio (UACR). The homeostasis model of assessment for insulin resistance (HOMA-IR) was calculated as [fasting blood glucose (FBG, mmol/L) × fasting blood insulin (FINS, μU/mL)]/22.5. The cutoff value of 2.5 was used to define insulin resistance (IR). The TyG index was calculated as Ln [Fasting Triglycerides (mg/dL) × Fasting Glucose (mg/dL)/2] [15]. Lp-PLA2 was measured using the enhanced immunoturbidimetric method, following the protocol provided by Medicalsystem Biotechnology Co., Ltd., Ningbo, China.

### 2.5. Liver Steatosis and Fibrosis Assessments

Ultrasonographic features evaluated using high-resolution B-mode ultrasonography (measured by two fixed and experienced investigators) for diagnosing steatotic liver disease include increased echogenicity of the liver parenchyma (“bright liver”), attenuation of the ultrasound beam in the far field, an enhancement in liver–kidney contrast, and unclear visualization of the intrahepatic structures. An unenhanced CT scan (Aquilion 64, Canon Medical Systems, with parameter settings as follows: tube current: 250 mA; tube voltage: 120 Kv; layer thickness: 5 mm) showing liver measurements of lower than 48 HU was considered indicative of liver steatosis [16,17]. Non-invasive fibrosis scores in the fibrosis-4 (FIB-4) index were computed with established cutoff values of >2.67 defining the presence of advance fibrosis [18].

### 2.6. Carotid Atherosclerosis (CAS) Evaluation

High-resolution B-mode ultrasonography was employed to assess carotid intima-media thickness (CIMT) and the presence of carotid plaques and conducted by two specialist sonographers with over ten years of experience.

CIMT was measured on the far wall of the common carotid artery. The specific site was approximately 1–2 cm proximal to the carotid bulb. Measurements were performed bilaterally (on both the left and right arteries). The mean value was calculated from all valid measurements averaged across both sides. We explicitly state that measurements were acquired in plaque-free segments, adhering to international consensus guidelines [19]. Intima-media thickening of the carotid artery is defined as a CIMT of ≥1.0 mm [17]. Carotid plaque is characterized as localized thickening of the arterial lumen greater than 0.5 mm or exceeding 50% of the surrounding intima-media thickness [20]. Carotid atherosclerosis was defined as intima-media thickening of the carotid artery or the presence of carotid plaque [21].

### 2.7. Statistical Methods

All research data were analyzed using R software (version 4.2.0). Normally distributed data are presented as the mean ± standard deviation, while non-normally distributed data are represented by the median (interquartile range). Comparisons between two groups of normally distributed continuous variables were conducted using *t*-tests, and multiple comparisons were performed using an analysis of variance with Bonferroni correction. Categorical variables are expressed as frequencies and percentages, with comparisons between two rates assessed using the chi-square test. Quartiles are statistical measures that divide a sorted dataset into four equal parts, each containing approximately 25% of the observations. Binary logistic regression analysis was employed to explore the relationship between different quartiles of Lp-PLA2 and both MASLD and CAS. Restricted cubic spline (RCS) regression was utilized to fit the nonlinear relationship between Lp-PLA2 and CAS. The determination of the number of nodes, set at three, was based on the minimum Akaike information criterion (AIC) value, with the 10th percentile (P10), 50th percentile (P50), and 90th percentile (P90) of Lp-PLA2 activity serving as the node positions. We performed a post hoc power analysis for the primary endpoint of our study, which was the association between elevated Lp-PLA2 activity (dichotomized at the 570 U/L threshold) and carotid atherosclerosis (CAS) in the high-FIB-4-score subgroup. The analysis was conducted using the observed results from our cohort. The analysis achieved a statistical power of 92.4% (α = 0.05, two-sided), which is well above the conventional threshold of 80%. This indicates that our study, with its final sample size of 1095 participants, was adequately powered to detect the significant association (OR = 2.67; 95% CI: 1.31–5.42; *p* = 0.007) we reported for this key finding in the specified high-risk subgroup. A significance level of *p* < 0.05 was considered indicative of statistical differences.

## 3. Results

### 3.1. Baseline Characteristics of MASLD

A total of 1095 participants (mean age: 54.8 years; 62.4% male) were included in the final analysis, among whom 69.8% had concomitant CAS (Figure 1). Baseline characteristics stratified by MASLD status are shown in Table 1. Patients with MASLD were younger (52.8 ± 12.9 vs. 58.1 ± 11.9 years, *p* < 0.001) and had a higher BMI, waist circumference, blood pressure, triglyceride–glucose index, estimated glomerular filtration rate, and metabolic syndrome prevalence and higher liver enzymes, uric acid, fasting plasma glucose, total cholesterol, and triglyceride levels compared to those in non-MASLD patients (all *p* < 0.001). No significant differences were observed in diabetes duration, sex distribution, or HbA1c levels. The non-MASLD group showed a higher prevalence of CAS (76.0% vs. 65.9%, *p* < 0.001), diabetic kidney disease, atherosclerotic cardiovascular disease, and carotid plaques (55.1% vs. 41.7%, *p* < 0.001) and a greater carotid intima-media thickness (1.11 ± 0.23 vs. 1.06 ± 0.24 mm, *p* < 0.001). Lp-PLA2 activity was significantly elevated in MASLD patients (573 ± 164 vs. 540 ± 170 U/L, *p* = 0.002). Regarding glucose-lowering medications, metformin use did not differ between groups, but non-MASLD patients had higher usage of insulin secretagogues (25.2% vs. 19.1%, *p* = 0.022) and insulin (21.6% vs. 12.5%, *p* < 0.001).

### 3.2. Lp-PLA2 Activity and MASLD Status

Given the elevated activity of Lp-PLA2 observed in the MASLD group, logistic regression was employed to clarify the association between Lp-PLA2 activity and the prevalence of MASLD (Appendix A). In an unadjusted model, a stepwise increase in Lp-PLA2 levels within the Q3 and Q4 cohorts was associated with a higher risk of MASLD (OR: 1.49, 95% CI: 1.05–2.10 for Q3; OR: 1.61, 95% CI: 1.14–2.28 for Q4). Moreover, an increase in Lp-PLA2 levels corresponded to an augmented risk of MASLD (*p* = 0.004). After adjusting for age and gender, only the Q4 group demonstrated a significantly increased risk of developing MASLD compared to the reference Q1 group (OR 1.47, 95% CI: 1.03–2.10, *p* = 0.025), with a consistent linear increase in MASLD risk across the groups. Subsequent adjustments for age, gender, and BMI resulted in attenuation of the dose–response relationships, with the OR values showing no statistically significant differences (OR 1.40, 95% CI: 1.03–1.90). The Q4 cohort exhibited elevated levels of CHOL, TGs, LDL-c, FBG, and HbA1c and an increased TyG index, alongside a higher prevalence of hypertension. Within the Q4 cohort, a greater percentage of individuals exhibited MASLD compared to that in the Q1 group (66.4% vs. 55.1%; Table 2).

In a sub-cohort of 249 cases of MASLD, liver fat estimation was performed using CT scans, with 186 participants classified as having mild hepatic steatosis and 63 as having moderate to severe fat accumulation. Notably, a comparison between these two groups revealed no statistically significant difference in Lp-PLA2 activity levels (611 ± 135 vs. 576 ± 177 U/L, *p* = 0.108, Appendix A). Furthermore, within the MASLD cohort, stratification based on FIB-4 levels indicated no statistically significant difference in Lp-PLA2 levels (578 ± 157 vs. 559 ± 179 U/L, *p* = 0.183, Appendix A). The corresponding scatter plot also demonstrated no significant linear relationship between Lp-PLA2 and FIB-4 (R = −0.053, *p* = 0.169, Appendix A). Regarding the relationship between Lp-PLA2 and lipid metabolism associated with MASLD (Figure 2), Lp-PLA2 exhibited positive correlations with CHOL, logarithmically transformed TGs, and LDL-c (Figure 2A–C).

### 3.3. Lp-PLA2 and Atherosclerosis in MASLD

In the analysis stratified by CAS status, 764 participants were classified into the CAS group (Appendix A). Based on previous evidence linking Lp-PLA2 levels to factors such as age, sex, and disease duration, propensity score matching was applied to minimize confounding and enable balanced comparisons between cohorts. Before matching, the non-CAS group had significantly higher values in several metabolic parameters—including CHOL, TGs, HDL-C, FBG, and HbA1c—while the CAS group showed greater use of statins, antiplatelet agents, metformin, insulin, and insulin secretagogues (*p* < 0.01). After matching, the baseline characteristics were well balanced (Appendix A). Notably, Lp-PLA2 activity was higher in the CAS group than that in the non-CAS group (563 ± 168 U/L vs. 554 ± 156 U/L, *p* = 0.53), although no significant differences were observed in the prevalence of MASLD, metabolic syndrome, or ASCVD.

To evaluate the association between Lp-PLA2 and CAS, we performed multivariable logistic regression adjusted for age, sex, disease duration, systolic blood pressure, smoking status, glycated hemoglobin, statin use, and diabetic kidney disease (Figure 3). No statistically significant odds ratios were observed across Lp-PLA2 quartiles in any of the adjusted models, and trend tests also indicated non-significance (Figure 3A). However, the restricted cubic spline (RCS) analysis revealed a nonlinear relationship between Lp-PLA2 activity and CAS, with distinct patterns emerging in subgroups with and without MASLD (Figure 3B). Further analysis of the dose–response relationship across FIB-4 strata showed that elevated Lp-PLA2 levels were associated with increased CAS risk beyond a threshold of 570 U/L, specifically in the high-FIB-4-score group, where each standard deviation increase in Lp-PLA2 corresponded to an odds ratio of 1.67 (95% CI: 1.12–2.48; Appendix A). In contrast, no significant association was detected in the low-FIB-4-score group (Figure 3C). These results underscore a context-dependent relationship between Lp-PLA2 and CAS, modulated by liver fibrosis severity and metabolic status (Figure 3D).

## 4. Discussion

Our study demonstrates that Lp-PLA2 activity is elevated in patients with T2DM and MASLD and is closely associated with metabolic risk factors, including dyslipidemia, insulin resistance, and metabolic syndrome. More importantly, we identified a nonlinear relationship between Lp-PLA2 and CAS, which was significantly modified by liver fibrosis severity. Specifically, Lp-PLA2 activity beyond a threshold of 570 U/L was associated with increased CAS risk only in those with advanced fibrosis, suggesting its potential role in cardiovascular risk stratification within this high-risk subgroup.

Consistent with previous biopsy-confirmed MASLD studies, we observed a higher Lp-PLA2 activity in T2DM patients with MASLD [20]. However, the drivers of Lp-PLA2 elevation in MASLD remain poorly understood. We further identified insulin resistance (IR) as a key driver of this elevation. As a precursor to type 2 diabetes (T2DM), IR contributes to the progression of both hepatic steatosis and atherosclerosis and may promote Lp-PLA2 synthesis through oxidative-stress-induced inflammation. This finding aligns with earlier reports in an Asian gestational diabetes cohort [22]. As a specific marker of vascular inflammation, Lp-PLA2 catalyzes the hydrolysis of oxidized phospholipids within LDL-c, generating two pro-inflammatory metabolites—lysophosphatidylcholine and oxidized free fatty acids—that promote IR in liver, muscle, and adipose tissues [9]. This interaction establishes a positive feedback loop wherein IR and Lp-PLA2 reciprocally amplify each other, underscoring the enzyme’s role as a pro-atherosclerotic factor in metabolic liver disease.

Although MASLD patients were younger and had better renal function and fewer diabetic complications at the baseline, the association between elevated Lp-PLA2 and CAS in the high-fibrosis subgroup remained independent after adjusting for these and other confounders. The higher CAS prevalence in non-MASLD patients likely reflects their older age and greater cumulative burden of traditional risk factors, underscoring the importance of interpreting Lp-PLA2 within the context of both metabolic liver disease and the overall stage of diabetes.

MASLD frequently coexists with T2DM, and the concurrent presence of these conditions significantly increases the risk of cardiovascular disease. It has been reported that Lp-PLA2 levels positively correlate with gamma-glutamyl transferase (GGT) and the fatty liver index (FLI) [23]. Our study found that advanced fibrosis, rather than the degree of steatosis, exhibited an inconsistent relationship between Lp-PLA2 and atherosclerosis when compared to individuals without advanced fibrosis. Nevertheless, the definitive causal relationship between Lp-PLA2, MASLD-associated fibrosis, and AS remains unclear. Previous animal model studies have demonstrated that upregulation of Lp-PLA2 expression in mice with high-fat-diet-induced MASLD promotes lipid and collagen accumulation by inhibiting autophagy and activating the JAK2/STAT3 signaling pathway [24]. Interestingly, the relationship between Lp-PLA2 and the FIB-4 index in this study did not exhibit a linear correlation but instead demonstrated a significant threshold effect. This phenomenon may be attributed to the diverse biological functions of Lp-PLA2, which are influenced by its distribution across different lipoproteins. Notably, the majority (70–80%) of Lp-PLA2 is associated with LDL-c, which is rich in apolipoprotein B, while a smaller fraction (approximately 20%) binds to HDL [25]. HDL-bound Lp-PLA2 is particularly considered to have anti-atherosclerotic properties [25]. Additionally, correlation analyses between Lp-PLA2 and lipid profiles indicated that irrespective of Lp-PLA2 levels, there was a positive correlation with HDL, especially prominent in the non-MASLD population [26]. Cirrhotic NASH patients exhibited significant differences, showing lower LDL-apoB production rates and reduced HDL-bound protein levels compared to those in their non-cirrhotic counterparts. The diminished LDL-apoB production may attenuate the effects of Lp-PLA2 when its activity is insufficient, thereby allowing the anti-atherosclerotic effects of HDL-Lp-PLA2 to overshadow the role of LDL-bound Lp-PLA2 as a risk factor for atherosclerosis at lower Lp-PLA2 activity levels, which may reduce its predictive value in atherosclerosis [27]. Furthermore, the MASLD group exhibited a clinical profile suggestive of an earlier-stage diabetic population, characterized by a significantly younger age, a higher eGFR, and a lower prevalence of diabetic and cardiovascular complications, despite a comparable formal duration of diabetes to that in the non-MASLD group. This suggests that MASLD may identify a subset of patients with T2DM who present with more pronounced metabolic dysfunction early in the disease course. Consequently, the higher prevalence of CAS in the non-MASLD group is likely attributable to their older age and greater cumulative burden of traditional risk factors over time. This underscores the importance of interpreting the role of Lp-PLA2 within the context of both metabolic liver disease and the overall stage of diabetes.

This study has several limitations. First, its cross-sectional and single-center design precludes causal inference. Second, hepatic steatosis was diagnosed through ultrasound, which may underdiagnose mild cases. Third, fibrosis was assessed non-invasively using the FIB-4 index, which has reduced accuracy in diabetic populations; future studies should employ more precise methods. Fourth, the MASLD cohort was significantly younger, potentially due to survivorship bias or more intensive screening in younger metabolic patients. Finally, the proposed mechanistic explanation regarding lipoprotein carriers remains speculative due to technical challenges in separating lipoprotein subfractions without affecting Lp-PLA2 activity. Prospective studies with larger sample sizes are needed to validate our findings and clarify the underlying mechanisms.

## 5. Conclusions

Individuals with T2DM and MASLD exhibit elevated Lp-PLA2 activity. However, Lp-PLA2 is not an independent predictor of MASLD onset, and its levels show no significant correlation with hepatic steatosis or fibrosis in this population. Instead, Lp-PLA2 is strongly associated with metabolic risk factors—including dyslipidemia, insulin resistance, and metabolic syndrome—among MASLD patients. Notably, in high-risk individuals with advanced fibrosis, Lp-PLA2 activity exceeding a threshold of 570 U/L is associated with CAS risk, highlighting its potential role in cardiovascular risk stratification within this subgroup. While carotid ultrasound remains a widely accessible and standard tool for assessing atherosclerosis, Lp-PLA2 may serve as an adjunct biomarker for identifying high-risk subgroups. Future studies should evaluate the cost effectiveness and clinical utility of integrating Lp-PLA2 into existing screening pathways.

## Figures and Tables

**Figure 1 biomedicines-13-02431-f001:**
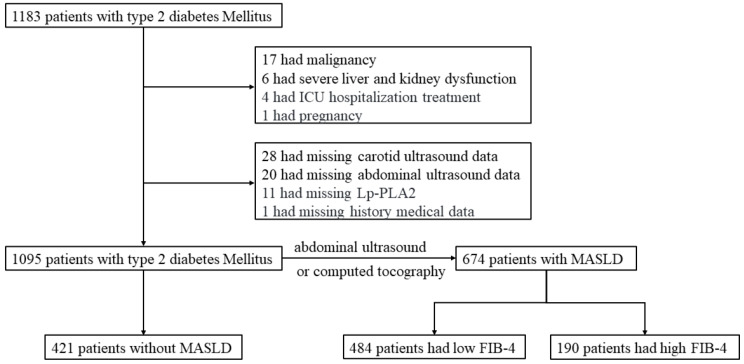
The flowchart of the present study. Abbreviation: ICU, Intensive Care Unit; MASLD, metabolic-dysfunction-associated steatotic liver disease; FIB-4, fibrosis-4.

**Figure 2 biomedicines-13-02431-f002:**
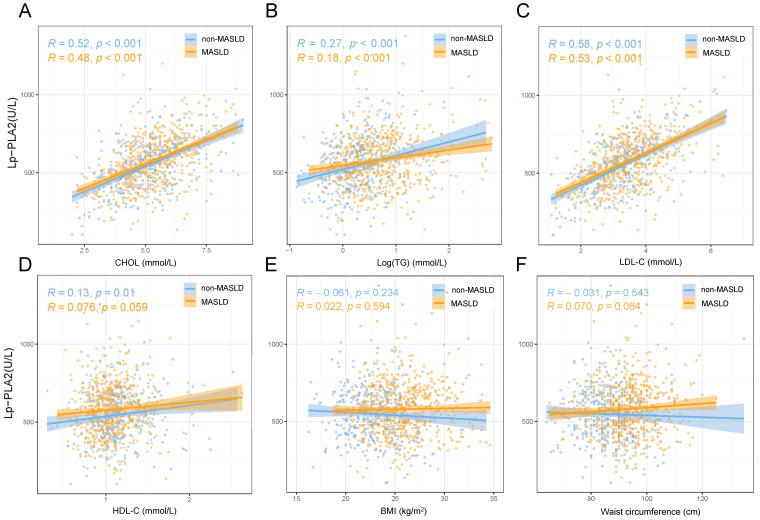
The relationship between Lp-PLA2 and lipid metabolism, body mass index, and waist circumference in metabolic-dysfunction-associated steatotic liver disease (MASLD). (**A**) Total cholesterol; (**B**) Log(triglycerides); (**C**) low-density lipoprotein cholesterol; (**D**) high-density lipoprotein cholesterol; (**E**) body mass index; (**F**) waist circumference.

**Figure 3 biomedicines-13-02431-f003:**
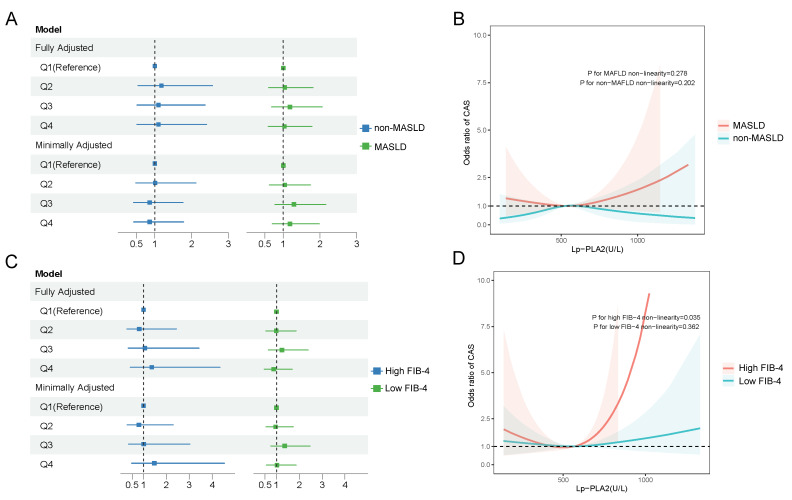
The relationship between Lp-PLA2 and carotid atherosclerosis (CAS). (**A**) Multivariable logistic regression model was utilized to adjust for confounding factors, including age, gender, disease duration, systolic blood pressure, smoking status, glycated hemoglobin levels, statin medication use, and diabetic kidney disease among MASLD patients and non-MASLD patients. “Minimally adjusted model” stands for adjusting age and sex. “Fully adjusted model” stands for adjusting age, sex, disease duration, systolic blood pressure, smoking, glycated hemoglobin, statin use, and diabetic nephropathy. (**B**) Restricted cubic spline (RCS) curves demonstrated a nonlinear relationship between Lp-PLA2 and CAS among MASLD patients and non-MASLD patients. (**C**) The multivariable logistic regression model was utilized to adjust for confounding factors, including age, gender, disease duration, systolic blood pressure, smoking status, glycated hemoglobin levels, statin medication use, and diabetic kidney disease, among MASLD patients with a high FIB-4 or low FIB-4 score. (**D**) “Minimally adjusted model” stands for adjusting age and sex. “Fully adjusted model” stands for adjusting age, sex, disease duration, systolic blood pressure, smoking, glycated hemoglobin, statin use, and diabetic nephropathy. (**B**) Restricted cubic spline (RCS) curves demonstrated a nonlinear relationship between Lp-PLA2 and CAS among MASLD patients with a high FIB-4 or a low FIB-4 score.

**Table 1 biomedicines-13-02431-t001:** Clinical characteristics of patients with metabolic-dysfunction-associated steatotic liver disease (MASLD) and without MASLD.

	Non-MASLD	MASLD	All	*p*
Characteristics	N = 421	N = 674	N = 1095	
Age (year)	58.1 ± 11.9	52.8 ± 12.9	54.8 ± 12.8	<0.001
Male (n, %)	272 (64.6)	411 (61.0)	683 (62.4)	0.25
Duration (month) ^†^	47 (11–91)	51 (14–97)	48(10–102)	0.43
Body mass index (kg/m^2^)	23.0 ± 2.9	25.9 ± 3.2	24.8 ± 3.4	<0.001
Waist circumstance (cm)	86.2 ± 8.7	93.4 ± 8.8	90.7 ± 9.4	<0.001
Hypertension (n, %)	162 (38.5)	266 (39.5)	428 (39.1)	0.79
Systolic blood pressure (mmHg)	131 ± 20	134 ± 19	132 ± 19	0.011
Diastolic blood pressure (mmHg)	79 ± 10	84 ± 11	82 ± 11	<0.001
Smoking (n, %)	137 (32.5)	184 (27.3)	321 (29.3)	0.07
Use of statins (n, %)	77 (18.3)	82 (12.2)	159 (14.5)	0.007
Use of antiplatelet drugs (n, %)	70 (16.6)	62 (9.2)	132 (12.1)	<0.001
Alanine aminotransferase (U/L) ^†^	18 (13–24)	27(19–37)	22(16–32)	<0.001
Aspartate aminotransferase (U/L) ^†^	19 (16–24)	22(18–29)	21(17–26)	<0.001
Uric acid (μmol/L)	343 ± 99	378 ± 104	364 ± 103	<0.001
eGFR (mL/min/1.73 m^2^)	90.9 ± 23.7	99.7 ± 18.1	96.3 ± 20.9	<0.001
Total cholesterol (mmol/L)	5.1 ± 1.5	5.3 ± 1.4	5.2 ± 1.5	0.011
Triglycerides (mmol/L) ^†^	1.3 (0.9–1.7)	1.8 (1.3–2.6)	1.6 (1.1–2.3)	<0.001
HDL-c (mmol/L)	1.2 ± 0.3	1.1 ± 0.3	1.1 ± 0.3	<0.001
LDL-c (mmol/L)	3.3 ± 1.1	3.4 ± 1.0	3.4 ± 1.0	0.008
Fasting blood glucose (mmol/L)	8.5 ± 3.6	9.2 ± 3.1	8.9 ± 3.3	0.001
HbA1c (%)	9.8 ± 2.6	9.9 ± 2.0	9.9 ± 2.3	0.24
TyG index	9.02 ± 0.75	9.50 ± 0.73	9.32 ± 0.77	<0.001
Lp-PLA2 (U/L)	540 ± 170	573 ± 164	560 ± 167	0.002
Use of diabetes drugs				
Metformin (n, %)	168 (39.9)	248 (36.8)	416 (38.0)	0.33
Insulin secretagogues (n, %)	106 (25.2)	129 (19.1)	235 (21.5)	0.022
Insulin (n, %)	91 (21.6)	84 (12.5)	175 (16.0)	<0.001
Metabolic syndrome (n, %)	216 (57.0)	511(83.4)	727(73.3)	<0.001
CIMT (cm)	1.11 ± 0.23	1.06 ± 0.24	1.08 ± 0.24	<0.001
Carotid plaque (n, %)	232 (55.1)	281 (41.7)	513 (46.8)	<0.001
Carotid atherosclerosis (n, %)	320 (76.0)	444 (65.9)	764 (69.8)	<0.001
DKD (n, %)	117 (27.8)	119 (17.7)	236 (21.6)	<0.001
ASCVD (n, %)	66 (15.7)	56 (8.3)	122 (11.1)	<0.001

^†^ Continuous variables are expressed as the median with the IQR for a non-Gaussian distribution. Abbreviation: MASLD, metabolic-dysfunction-associated steatotic liver disease; eGFR, estimated glomerular filtration rate; HDL-c, high-density lipoprotein-cholesterol; LDL-c, low-density lipoprotein-cholesterol; TyG index, triglyceride–glucose index; Lp-PLA2, lipoprotein-associated phospholipase A2; CIMT, carotid intima-media thickness; DKD, diabetic kidney disease; ASCVD, atherosclerotic cardiovascular disease.

**Table 2 biomedicines-13-02431-t002:** Clinical characteristics of all patients grouped by Q1–Q4 of Lp-PLA2.

	Q1 [96,450]	Q2 (450,556]	Q3 (556,666]	Q4 (666,1380]	*p*
	N = 276	N = 277	N = 268	N = 274	
Age (year)	56.9 ± 12.6 ^d^	56.1 ± 12.5 ^d^	54.2 ± 12.3	52.2 ± 13.3	<0.001
Male (n, %)	157 (56.9) ^d^	168 (60.6)	164 (61.2)	194 (70.8)	0.006
Duration (month) ^†^	78 (12–156) ^cd^	60 (12–120) ^cd^	36(6–108) ^d^	24 (2–72)	<0.001
Body mass index (kg/m^2^)	24.8 ± 3.5	24.7 ± 3.2	24.6 ± 3.2	25.1 ± 3.7	0.29
Waist circumstance (cm)	90.7 ± 9.5	90.1 ± 8.8	90.1 ± 9.5	91.7 ± 9.9	0.15
Hypertension (n, %)	133 (48.2) ^cd^	118 (42.6)	90 (33.6)	87 (31.8)	<0.001
Systolic blood pressure (mmHg)	132 ± 19	132 ± 20	133 ± 19	132 ± 20	0.82
Diastolic blood pressure (mmHg)	81 ± 11	81 ± 11	83 ± 11	83 ± 11	0.09
Smoking (n, %)	74 (26.8)	81 (29.2)	71 (26.5)	95 (34.7)	0.12
Use of statins (n, %)	78 (28.3) ^bcd^	49 (17.7) ^cd^	20 (7.5)	12 (4.4)	<0.001
Use of antiplatelet drugs (n, %)	52 (18.8) ^cd^	44 (15.9) ^d^	24 (9.0)	12 (4.4)	<0.001
Alanine aminotransferase (U/L) ^†^	21 (14–29) ^d^	22 (16–30) ^d^	22 (16–32) ^d^	26 (18–39)	<0.001
Aspartate aminotransferase (U/L) ^†^	20 (17–26)	21 (17–25)	20 (17–26)	22 (18–30)	0.06
Uric acid (μmol/L)	361 ± 101	360 ± 103	361 ± 99	375 ± 109	0.28
eGFR (mL/min/1.73 m^2^)	92.7 ± 20.7 ^cd^	94.9 ± 21.1	98.8 ± 20.0	99.0 ± 21.2	<0.001
Total cholesterol (mmol/L)	4.41 ± 1.20 ^bcd^	4.79 ± 1.11 ^cd^	5.52 ± 1.05 ^d^	6.21 ± 1.67	<0.001
Triglycerides (mmol/L) ^†^	1.40 (1.00–2.01) ^cd^	1.44 (1.06–2.04) ^cd^	1.57 (1.18–2.44)	1.81 (1.34–2.57)	<0.001
HDL-c (mmol/L)	1.08 ± 0.29	1.10 ± 0.27	1.11 ± 0.24	1.14 ± 0.31	0.09
LDL-c (mmol/L)	2.78 ± 0.86 ^bcd^	3.03 ± 0.79 ^cd^	3.59 ± 0.69 ^d^	4.07 ± 1.11	<0.001
Fasting blood glucose (mmol/L)	8.4 ± 3.3 ^cd^	8.3 ± 3.1 ^cd^	9.5 ± 3.3	9.6 ± 3.4	<0.001
HbA1c (%)	9.6 ± 2.3 ^d^	9.6 ± 2.3 ^d^	10.1 ± 2.2	10.4 ± 2.2	<0.001
TyG index	9.11 ± 0.75 ^cd^	9.16 ± 0.78 ^cd^	9.44 ± 0.75	9.56 ± 0.73	<0.001
Lp-PLA2 (U/L)	359 ± 74 ^bcd^	504 ± 30 ^cd^	609 ± 32 ^d^	771 ± 114	<0.001
Metabolic syndrome (n, %)	168 (71.5)	174 (71.0)	183 (73.2)	202 (77.1)	0.39
MASLD (n, %)	152 (55.1) ^d^	167 (60.3)	173 (64.6)	182 (66.4)	0.031
CIMT (cm)	1.09 ± 0.24	1.08 ± 0.24	1.09 ± 0.24	1.08 ± 0.24	0.87
Carotid plaque (n, %)	140 (50.7)	122 (44.0)	132 (49.3)	119 (43.4)	0.21
CAS [n, (%)]	200 (72.5)	191 (69.0)	189 (70.5)	184 (67.2)	0.57
DKD (n, %)	78 (28.3%) ^c^	54 (19.5%)	46 (17.2%)	58 (21.2%)	0.011
ASCVD (n, %)	43 (15.6) ^d^	35 (12.6)	23 (8.6)	21 (7.7)	0.011

Q1, Q2, Q3, and Q4 correspond to the first, second, third, and fourth quartiles of Lp-PLA2 levels. Pairwise comparisons between groups: ^b^ indicates *p* < 0.05 compared to the Q2 group, ^c^ indicates *p* < 0.05 compared to the Q3 group, and ^d^ indicates *p* < 0.05 compared to the Q4 group. ^†^ Continuous variables are expressed as the median with the IQR for a non-Gaussian distribution. Abbreviations: MASLD, metabolic-dysfunction-associated steatotic liver disease; eGFR, estimated glomerular filtration rate; HDL-c, high-density lipoprotein-cholesterol; TyG index, triglyceride–glucose index; Lp-PLA2, lipoprotein-associated phospholipase A2; CIMT, carotid intima-media thickness; DKD, diabetic kidney disease; ASCVD, atherosclerotic cardiovascular disease.

## Data Availability

The datasets used and/or analyzed during the current study are available from the corresponding author on reasonable request.

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
