# Peer review of "Fibrosis Severity in MASLD Determines the Predictive Value of Lp-PLA2 for Carotid Atherosclerosis in Type 2 Diabetes: A Cross-Sectional Study"

_biomedicines, 2025, doi:10.3390/biomedicines13102431_

Round 1

Reviewer 1 Report

Comments and Suggestions for Authors

The study addresses a clinically interesting topic, and particularly the finding regarding the “Lp-PLA2 threshold in the advanced fibrosis MASLD subgroup” may be valuable. However, methodological limitations and contradictory aspects of the results (likely due to age differences, especially the higher prevalence of CAS in non-MASLD patients) diminish the scientific impact of the study. The results should be interpreted with greater caution, and the limitations need to be emphasized more strongly. Moreover, when reading the entire paper, although it addresses an important topic, the abstract and results sections in particular make it unclear what exactly is being studied, what was actually found, and how the findings should be interpreted. In its current form, I believe readers will need to put in extra effort to understand the manuscript. Therefore, the paper needs to be revised throughout, both in terms of the presentation and interpretation of the results and overall readability. In addition, below I provide specific comments that I think should be improved for each section:

Title: Too long and complex; it does not immediately convey the “main message” to the reader.
The phrase “Steatotic liver fibrosis modifies the effects…” is abstract and complicated. It is difficult to understand exactly in which patients, under which conditions, what is being modified. From what I understand, the intended meaning is: “In T2DM patients with MASLD + fibrosis, the predictive power of Lp-PLA2 for atherosclerosis changes.” Therefore, I suggest revising the title as:
“Fibrosis Severity in MASLD Determines the Predictive Value of Lp-PLA2 for Carotid Atherosclerosis in Type 2 Diabetes.”

Also, the phrase “observational study” does not fully reflect this manuscript. Although the patients are presented as if “prospectively enrolled,” in reality all measurements were made at a single time point. Therefore, the study is cross-sectional.
The term “observational study” gives the impression of a longitudinal cohort, which would be misleading.
Thus, the phrase should be changed to “cross-sectional study” in the title. Otherwise, readers may misinterpret the study design.

Abstract: The methods and results should be rewritten more clearly and transparently. From this section, it is difficult to understand the main findings without reading the full manuscript. Also, an OR is incorrectly reported as 2.67 instead of 1.67. The lack of association between steatosis degree and Lp-PLA2 is an important negative finding. Therefore, this should be emphasized in the Conclusion, and the fact that only in the advanced fibrosis MASLD subgroup the Lp-PLA2 threshold may be of potential use—but within the limitations of a cross-sectional study—must be highlighted.

Introduction: The MASLD–T2DM–CAS context is not well established. The interrelationship of these three pathologies should be better contextualized, and the hypothesis/aim of the study presented in a clearer and more straightforward way.

Materials and Methods: The description of excessive alcohol consumption as an exclusion criterion is vague: what amount, what duration? A reference is needed.

Although the authors state that CIMT was measured by experienced sonographers with repeated measurements, several important methodological details are missing. The manuscript does not specify which carotid segment, which side of the artery, or which wall (posterior/anterior) was measured, nor whether mean values from both carotid arteries were averaged. In addition, international consensus statements recommend that CIMT be measured in plaque-free segments of the common carotid artery to avoid overestimation and variability. The authors did not clarify whether this principle was followed, which raises concerns about the accuracy of their measurements and comparability with established CIMT studies.

Results: The presentation of the results is confusing rather than focused. Data from the tables are repeated verbatim. It is unclear which findings are truly important and which are secondary. The large number of results makes it difficult to follow the main message. Group differences are presented, but in such a heterogeneous population (MASLD + DM + varying age + statin use, etc.), the clinical significance of these differences is not sufficiently discussed. The discussion section also does not interpret the results adequately.

Some findings are nevertheless interesting:

  1. MASLD patients being younger: Contrary to expectations, MASLD is generally more common with increasing age. This reversed association may be due to selection bias (e.g., older MASLD patients being lost earlier due to diabetic complications, i.e., survivorship bias). Also, younger MASLD patients may be under closer follow-up.
  2. Higher statin use in MASLD patients: MASLD often coexists with dyslipidemia, so statin use is expected. However, statins also have anti-inflammatory effects, which may mask CIMT findings or inflammatory markers. This is an important confounder.
  3. Higher eGFR and lower diabetic nephropathy in MASLD patients: This is likely age-related, as the MASLD group is younger, which protects kidney function.

Although diabetes duration was not statistically different, overall the MASLD group appears to represent an “earlier-stage diabetic” population. The authors did not address these issues sufficiently in the discussion.

Discussion: This section is quite superficial, with insufficient interpretation of the results in light of the literature. The clinical implications are not adequately discussed. The limitations mentioned overlap with what I also consider limitations, but key issues—single time-point measurement, MASLD group being younger, heterogeneity of the patient population, and limitations of fibrosis and CAS assessment methods—should be more strongly emphasized. The findings should be presented more as preliminary.

Conclusion: This is a cross-sectional, single-center study with a limited sample size. The conclusions should therefore be hypothesis-generating rather than prescriptive. Suggesting a clinical algorithm (“first measure Lp-PLA2, then decide on USG if needed”) goes beyond the strength of the evidence. The authors also present Lp-PLA2 as an alternative screening test to carotid ultrasound. However, Lp-PLA2 measurement is a commercial, costly assay that is not widely available, while carotid ultrasound is already a routine or low-cost procedure in many centers, particularly for diabetes/liver patients. The authors did not compare the cost of Lp-PLA2 with carotid ultrasound, making claims of being “more practical” or “cost-effective” unfounded.

References: For such a current topic, the references are quite outdated. They need to be substantially updated.

Comments on the Quality of English Language

The English is generally understandable and meets academic standards. However, the text frequently uses long, complex, and complex sentences. This reduces fluidity and readability. Sentences, particularly in the abstract and results sections, are very convoluted, making it difficult for the reader to grasp the main message.

Author Response

Dear Reviewer,

We are most grateful for the time and care the editor and reviewers spent for improving our manuscript.

Please find below the point-by-point responses to your comments. All revisions have been highlighted with a yellow background for clarity.

Sincerely,

Congxiang Shao and Bihui Zhong

On behalf of the author team

Reviewer 1

Comments and Suggestions for Authors

The study addresses a clinically interesting topic, and particularly the finding regarding the “Lp-PLA2 threshold in the advanced fibrosis MASLD subgroup” may be valuable. However, methodological limitations and contradictory aspects of the results (likely due to age differences, especially the higher prevalence of CAS in non-MASLD patients) diminish the scientific impact of the study. The results should be interpreted with greater caution, and the limitations need to be emphasized more strongly. Moreover, when reading the entire paper, although it addresses an important topic, the abstract and results sections in particular make it unclear what exactly is being studied, what was actually found, and how the findings should be interpreted. In its current form, I believe readers will need to put in extra effort to understand the manuscript. Therefore, the paper needs to be revised throughout, both in terms of the presentation and interpretation of the results and overall readability. In addition, below I provide specific comments that I think should be improved for each section:

Title: Too long and complex; it does not immediately convey the “main message” to the reader.

The phrase “Steatotic liver fibrosis modifies the effects…” is abstract and complicated. It is difficult to understand exactly in which patients, under which conditions, what is being modified. From what I understand, the intended meaning is: “In T2DM patients with MASLD + fibrosis, the predictive power of Lp-PLA2 for atherosclerosis changes.” Therefore, I suggest revising the title as: “Fibrosis Severity in MASLD Determines the Predictive Value of Lp-PLA2 for Carotid Atherosclerosis in Type 2 Diabetes.”

Reply: Thanks for the enlightening suggestion. As you suggested, we have revised the paper title to "Fibrosis Severity in MASLD Determines the Predictive Value of Lp-PLA2 for Carotid Atherosclerosis in Type 2 Diabetes".

Also, the phrase “observational study” does not fully reflect this manuscript. Although the patients are presented as if “prospectively enrolled,” in reality all measurements were made at a single time point. Therefore, the study is cross-sectional.

The term “observational study” gives the impression of a longitudinal cohort, which would be misleading.

Thus, the phrase should be changed to “cross-sectional study” in the title. Otherwise, readers may misinterpret the study design.

Reply: Thank you for your valuable suggestion. Based on your suggestion, we have revised the term "observational study" to "cross-sectional study".

Abstract: The methods and results should be rewritten more clearly and transparently. From this section, it is difficult to understand the main findings without reading the full manuscript. Also, an OR is incorrectly reported as 2.67 instead of 1.67. The lack of association between steatosis degree and Lp-PLA2 is an important negative finding. Therefore, this should be emphasized in the Conclusion, and the fact that only in the advanced fibrosis MASLD subgroup the Lp-PLA2 threshold may be of potential use—but within the limitations of a cross-sectional study—must be highlighted.

Reply: We feel so sorry for this mistake and we have rectified the whole abstract as you suggested and the revised abstract also meets the word count requirement.

Introduction: The MASLD–T2DM–CAS context is not well established. The interrelationship of these three pathologies should be better contextualized, and the hypothesis/aim of the study presented in a clearer and more straightforward way.

Reply: Thanks for the enlightening suggestion. We have rewritten the introduction section. as follows:

Metabolic dysfunction-associated steatotic liver disease (MASLD) encompasses a spectrum of liver conditions induced by metabolic stress, ranging from simple steatosis to steatohepatitis, cirrhosis, and hepatocellular carcinoma. MASLD is intrinsically linked to metabolic syndrome and is a significant contributor to extrahepatic complications, particularly type 2 diabetes mellitus (T2DM) and cardiovascular disease (CVD). The coexistence of MASLD and T2DM is highly prevalent and creates a synergistic adverse effect, accelerating the progression of both liver and cardiovascular damage. This interplay is driven by shared pathophysiological mechanisms, including insulin resistance, chronic inflammation, and dyslipidemia, which collectively promote the development of atherosclerosis—a key feature of CVD.

Atherosclerosis, particularly in the carotid arteries (carotid atherosclerosis, CAS), is a major cause of morbidity and mortality in patients with T2DM. The disease is marked by abnormalities in serum lipids and vascular inflammation. Lipoprotein-associated phospholipase A2 (Lp-PLA2) serves as a vascular-specific inflammatory marker that plays a significant role in the progression of atherosclerosis. As an enzyme that catalyzes the hydrolysis of oxidized phospholipids on the surface of LDL particles, Lp-PLA2 produces pro-inflammatory mediators, such as lysophosphatidylcholine and oxidized nonesterified fatty acids, positioning it as an independent predictor of coronary artery disease.

The liver, a primary organ responsible for metabolic homeostasis, also plays a crucial role in the clearance of circulating Lp-PLA2. Importantly, the ability of Lp-PLA2 to predict CVD in individuals with T2DM appears to be influenced by the presence of concomitant metabolic disorders, such as MASLD. However, previous studies have reported conflicting results regarding changes in Lp-PLA2 concentrations in MASLD, and it remains unclear whether Lp-PLA2 activity, which more accurately reflects its biological role, is affected by MASLD status. Moreover, whether MASLD, especially its more advanced fibrotic stages, modifies the association between Lp-PLA2 and atherosclerosis in the high-risk T2DM population is unknown.

Therefore, we aimed to investigate the interrelationship between Lp-PLA2 activity, MASLD, and CAS in patients with T2DM. Besides, the primary aim of this study was to investigate whether Lp-PLA2 activity predicts carotid atherosclerosis in patients with T2DM and MASLD, and to determine if this relationship is modified by the severity of liver fibrosis.

Materials and Methods: The description of excessive alcohol consumption as an exclusion criterion is vague: what amount, what duration? A reference is needed.

Reply: Thanks for the enlightening suggestion. We have provided a detailed explanation of excessive alcohol intake and included references.

Significant alcohol consumption was defined as an average intake of >30 g/day for men and >20 g/day for women over the 3 months prior to study inclusion. (Line 96 to 98)

Reference:

[1] Torp N, Israelsen M, Johansen S, Semmler G, Dalby Hansen C, Bech KT, et al. MetALD: Diagnosis and Prognosis With Non-Invasive Tests. Aliment Pharmacol Ther. 2025 Aug 11. Epub ahead of print.

[2] European Association for the Study of the Liver (EASL); European Association for the Study of Diabetes (EASD); European Association for the Study of Obesity (EASO). EASL-EASD-EASO Clinical Practice Guidelines on the management of metabolic dysfunction-associated steatotic liver disease (MASLD). J Hepatol. 2024; 81:492-542.

Although the authors state that CIMT was measured by experienced sonographers with repeated measurements, several important methodological details are missing. The manuscript does not specify which carotid segment, which side of the artery, or which wall (posterior/anterior) was measured, nor whether mean values from both carotid arteries were averaged. In addition, international consensus statements recommend that CIMT be measured in plaque-free segments of the common carotid artery to avoid overestimation and variability. The authors did not clarify whether this principle was followed, which raises concerns about the accuracy of their measurements and comparability with established CIMT studies.

Reply: We thank the reviewer for raising this important point. We have added the detailed process as follows: (Line 158 to 167)

Measurements were taken on the far wall of the common carotid artery (CCA). The specific site was approximately 1-2 cm proximal to the carotid bulb. Measurements were performed bilaterally (on both the left and right arteries). The mean value was calculated from all valid measurements averaged across both sides. We explicitly state that measurements were acquired in plaque-free segments, adhering to international consensus guidelines.

Reference

[1] Touboul PJ, Hennerici MG, Meairs S, Adams H, Amarenco P, Bornstein N, et al. Mannheim carotid intima-media thickness and plaque consensus (2004-2006-2011). An update on behalf of the advisory board of the 3rd, 4th and 5th watching the risk symposia, at the 13th, 15th and 20th European Stroke Conferences, Mannheim, Germany, 2004, Brussels, Belgium, 2006, and Hamburg, Germany, 2011. Cerebrovasc Dis. 2012; 34:290-296.

[2] Stein JH, Korcarz CE, Hurst RT, Lonn E, Kendall CB, Mohler ER, et al. Use of carotid ultrasound to identify subclinical vascular disease and evaluate cardiovascular disease risk: a consensus statement from the American Society of Echocardiography Carotid Intima-Media Thickness Task Force. Endorsed by the Society for Vascular Medicine. J Am Soc Echocardiogr. 2008;21(2):93-111.

Results: The presentation of the results is confusing rather than focused. Data from the tables are repeated verbatim. It is unclear which findings are truly important and which are secondary. The large number of results makes it difficult to follow the main message. Group differences are presented, but in such a heterogeneous population (MASLD + DM + varying age + statin use, etc.), the clinical significance of these differences is not sufficiently discussed. The discussion section also does not interpret the results adequately.

Reply: We thank the reviewer for the enlightening suggestion and we have rewritten the result section.

Some findings are nevertheless interesting:

MASLD patients being younger: Contrary to expectations, MASLD is generally more common with increasing age. This reversed association may be due to selection bias (e.g., older MASLD patients being lost earlier due to diabetic complications, i.e., survivorship bias). Also, younger MASLD patients may be under closer follow-up.

Reply: We appreciate the reviewer’s insightful observation. Indeed, the prevalence of MASLD typically increases with age. However, in our study, the MASLD group was significantly younger than the non-MASLD group (52.8±12.9 vs. 58.1±11.9 years, p<0.001). We agree that this finding may reflect a form of selection bias, particularly survivorship bias, as older patients with both T2DM and MASLD may have already developed severe diabetic complications (e.g., cardiovascular disease, advanced liver fibrosis, or renal impairment) and thus were less likely to be included in our study cohort. Additionally, younger MASLD patients with T2DM are more likely to be under closer medical supervision due to their metabolic risk profiles, leading to earlier diagnosis and inclusion.

To address this concern, we have added the following clarification in the limitation part of Discussion section: (Line 386 to 391)

"Our MASLD cohort was significantly younger than the non-MASLD group. This may be attributable to survivorship bias, wherein older patients with both T2DM and MASLD are more likely to have advanced complications and thus were underrepresented in our study. Alternatively, younger patients with metabolic risk factors may be more frequently screened and diagnosed with MASLD, leading to their overrepresentation in clinical settings."

Higher statin use in MASLD patients: MASLD often coexists with dyslipidemia, so statin use is expected. However, statins also have anti-inflammatory effects, which may mask CIMT findings or inflammatory markers. This is an important confounder.

Reply: We thank the reviewer for raising this important point. It is true that statins possess anti-inflammatory and pleiotropic effects that could potentially confound the relationship between Lp-PLA2 and carotid atherosclerosis.

In our study, statin use was lower in the MASLD group compared to the non-MASLD group (12.2% vs. 18.3%, p=0.007). This may reflect under-prescribing in MASLD patients due to concerns about liver enzyme elevations, despite current guidelines supporting statin use in MASLD.

To further control for the potential confounding effect of statins, we already included statin use as a covariate in our multivariable logistic regression models when assessing the association between Lp-PLA2 and carotid atherosclerosis (as described in figure legend of Figure 3). This adjustment helps mitigate the influence of statins on inflammatory and vascular outcomes.

Higher eGFR and lower diabetic nephropathy in MASLD patients: This is likely age-related, as the MASLD group is younger, which protects kidney function.

Reply: We sincerely thank the reviewer for this astute observation. We completely agree with the reviewer that the significantly younger age of the MASLD cohort (52.8 ± 12.9 vs. 58.1 ± 11.9 years, p<0.001) is the most plausible explanation for the observed higher estimated glomerular filtration rate (eGFR) and lower prevalence of diabetic kidney disease (DKD) in this group, as age is a well-established, primary determinant of renal function.

To ensure that our analysis and interpretation of the relationship between Lp-PLA2 and carotid atherosclerosis (CAS) were not confounded by this or other baseline differences (such as age, kidney function, and medication use), we employed rigorous statistical adjustments: In our primary multivariable logistic regression models (Figure 3 of the manuscript), we adjusted for age, diabetic kidney disease (DKD), and statin use, among other covariates, when assessing the independent association between Lp-PLA2 and CAS. This approach directly controls for the confounding effect of age and renal status on the outcome. Furthermore, we utilized propensity score matching (mentioned in Section 3.3 and detailed in Supplementary Table 3) to create comparable groups between CAS and non-CAS patients, balancing characteristics including age and metabolic parameters, before comparing their Lp-PLA2 levels.

Therefore, while the baseline table correctly describes the raw differences between the groups, our core findings regarding the threshold effect of Lp-PLA2 on CAS risk in the high FIB-4 subgroup are based on models that have statistically accounted for these demographic and clinical disparities, including age and renal function. We have now revised the Discussion section to explicitly acknowledge this point and clarify our interpretation: (Line 330 to 336)

"At baseline, patients with MASLD were significantly younger and exhibited higher eGFR and a lower prevalence of DKD compared to non-MASLD patients. This likely reflects an age-related protective effect on kidney function in the MASLD group rather than a direct renal-protective effect of MASLD itself. Nevertheless, our primary analysis of the association between Lp-PLA2 and CAS employed multivariable regression models adjusted for age, DKD, and other potential confounders, ensuring that the identified threshold effect of Lp-PLA2 is independent of these baseline differences."

Although diabetes duration was not statistically different, overall the MASLD group appears to represent an “earlier-stage diabetic” population. The authors did not address these issues sufficiently in the discussion.

Reply: We thank the reviewer for raising this critical point. We agree that the collective baseline characteristics, specifically the significantly younger age, better renal function (higher eGFR), and lower prevalence of diabetic (DKD) and atherosclerotic (ASCVD, carotid plaque) complications-paint a picture of the MASLD cohort as representing a phenotypically "earlier-stage" or less advanced diabetic population, even in the absence of a statistically significant difference in the recorded duration of diabetes.

This is a crucial observation because it suggests that MASLD may develop or be diagnosed earlier in the natural history of T2DM, potentially acting as a marker of a more severe metabolic phenotype (evidenced by higher BMI, worse lipid profiles, and higher prevalence of metabolic syndrome) that manifests at a younger age. The higher prevalence of CAS in the non-MASLD group is therefore likely driven by their older age and longer exposure to other cumulative risk factors, rather than the absence of MASLD itself.

We have revised the Discussion section to explicitly acknowledge and discuss this concept: (Line 364 to 373)
"Furthermore, the MASLD group exhibited a clinical profile suggestive of an earlier-stage diabetic population, characterized by significantly younger age, higher eGFR, and lower prevalence of diabetic and cardiovascular complications, despite a comparable formal duration of diabetes to the non-MASLD group. This suggests that MASLD may identify a subset of patients with T2DM who present with a more pronounced metabolic dysfunction early in the disease course. Consequently, the higher prevalence of CAS in the non-MASLD group is likely attributable to their older age and greater cumulative burden of traditional risk factors over time. This underscores the importance of interpreting the role of Lp-PLA2 within the context of both metabolic liver disease and the overall stage of diabetes."

Discussion: This section is quite superficial, with insufficient interpretation of the results in light of the literature. The clinical implications are not adequately discussed. The limitations mentioned overlap with what I also consider limitations, but key issues—single time-point measurement, MASLD group being younger, heterogeneity of the patient population, and limitations of fibrosis and CAS assessment methods—should be more strongly emphasized. The findings should be presented more as preliminary.

Reply: We thank the reviewer for raising this important point. All the issues mentioned above have been addressed in accordance with the revision feedback above.

Conclusion: This is a cross-sectional, single-center study with a limited sample size. The conclusions should therefore be hypothesis-generating rather than prescriptive. Suggesting a clinical algorithm (“first measure Lp-PLA2, then decide on USG if needed”) goes beyond the strength of the evidence. The authors also present Lp-PLA2 as an alternative screening test to carotid ultrasound. However, Lp-PLA2 measurement is a commercial, costly assay that is not widely available, while carotid ultrasound is already a routine or low-cost procedure in many centers, particularly for diabetes/liver patients. The authors did not compare the cost of Lp-PLA2 with carotid ultrasound, making claims of being “more practical” or “cost-effective” unfounded.

Reply: Thank you for your valuable suggestion. We have rectified the Conclusion as you suggested. (Line 400 to 410)

Individuals with T2DM and MASLD exhibit elevated Lp-PLA2 activity. However, Lp-PLA2 is not an independent predictor of MASLD onset, and its levels show no sig-nificant correlation with hepatic steatosis or fibrosis in this population. Instead, Lp-PLA2 is strongly associated with metabolic risk factors—including dyslipidemia, insulin resistance, and metabolic syndrome—among MASLD patients. Notably, in high-risk individuals with advanced fibrosis, Lp-PLA2 activity exceeding a threshold of 570 U/L predicts CAS, highlighting its potential role in cardiovascular risk stratification within this subgroup. While carotid ultrasound remains a widely accessible and standard tool for assessing atherosclerosis, Lp-PLA2 may serve as an adjunct biomarker to identify high-risk subgroups. Future studies should evaluate the cost-effectiveness and clinical utility of integrating Lp-PLA2 into existing screening pathways.

References: For such a current topic, the references are quite outdated. They need to be substantially updated.

Reply: Thank you for your kind reminding. We have updated the reference.

Comments on the Quality of English Language

The English is generally understandable and meets academic standards. However, the text frequently uses long, complex, and complex sentences. This reduces fluidity and readability. Sentences, particularly in the abstract and results sections, are very convoluted, making it difficult for the reader to grasp the main message.

Reply: Thank you for your suggestion. We have improved the quality of English expression.

Reviewer 2 Report

Comments and Suggestions for Authors

Dear Authors,
I have undertaken the review of the manuscript on the effect of Lp-PLA2 on cardiovascular risk in patients with T2DM and MASLD with great interest. The manuscript is well-prepared, but it requires some clarifications and possible modifications.
I am attaching my comments.
1. Several doubts arise in the abstract. Here, in the description of the results and discussion, you mention insulin resistance – how did you diagnose it? I understand that the study included only patients with T2DM. Did all participants have insulin resistance? How did you diagnose it?
2. In Figure 1, which presents the study outline, you indicated ultrasound as the only method for diagnosing fatty liver disease – and what about computed tocography – it was also used, as I understand it. 3. What were the FIB-4 index values ​​in the compared groups of patients with and without MASLD? Some of these patients had an elevated FIB-4 index (n=190), indicating a fibrotic process, which, regardless of other factors, can increase cardiovascular risk. This situation may confound your results.
4. The results include cohorts Q1-Q4. The methods section should provide a more detailed description of their origins (to facilitate reader understanding of the results). The same applies to the TyG index.
5. The results include "Carotid atherosclerosis (n, %)." What is this parameter? How was it defined? Based on CIMT and carotid plaque? I'm wondering if someone had a high CIMT value and a low carotid plaque (or vice versa), did you also recognize CAS? This needs to be clearly defined.
In my opinion, the comments presented require clarification to ensure the presented results are clear. I will be happy to get to know them.

Author Response

Dear Reviewer,

We are most grateful for the time and care the editor and reviewers spent for improving our manuscript.

Please find below the point-by-point responses to your comments. All revisions have been highlighted with a yellow background for clarity.

Sincerely,

Congxiang Shao and Bihui Zhong,

On behalf of the author team

Reviewer 2

Comments and Suggestions for Authors

Dear Authors,
I have undertaken the review of the manuscript on the effect of Lp-PLA2 on cardiovascular risk in patients with T2DM and MASLD with great interest. The manuscript is well-prepared, but it requires some clarifications and possible modifications. I am attaching my comments.
1. Several doubts arise in the abstract. Here, in the description of the results and discussion, you mention insulin resistance – how did you diagnose it? I understand that the study included only patients with T2DM. Did all participants have insulin resistance? How did you diagnose it?

Response: We thank the reviewer for the kind reminding. The homeostasis model of assessment for insulin resistance (HOMA-IR) was calculated as [fasting blood glucose (FBG, mmol/L) × fasting blood insulin (FINS, μU/ml)]/22.5. The cutoff value of 2.5 was used to define insulin resistance (IR). And we had added it in the method section. (Line 137 to 141)

  1. In Figure 1, which presents the study outline, you indicated ultrasound as the only method for diagnosing fatty liver disease–and what about computed tocography–it was also used, as I understand it.

Response: Thank you for your thoughtful reminder. In the present study, liver steatosis was assessed via abdominal ultrasound or computed tomography. We have rectified the Figure 1.

  1. What were the FIB-4 index values ​​in the compared groups of patients with and without MASLD? Some of these patients had an elevated FIB-4 index (n=190), indicating a fibrotic process, which, regardless of other factors, can increase cardiovascular risk. This situation may confound your results.

Response: Thank you for your valuable suggestion. The patients with MASLD presented higher level of FIB-4 than the ones without MASLD. An elevated FIB-4 index indicating a fibrotic process, which, regardless of other factors, can increase cardiovascular risk and confound our results. Future studies with larger sample sizes and prospective designs are warranted to validate these results. We have added this in the limitation part. (Line 391 to 394)

  1. The results include cohorts Q1-Q4. The methods section should provide a more detailed description of their origins (to facilitate reader understanding of the results). The same applies to the TyG index.

Response: Thank you for your thoughtful reminder. We have added the relevant description: “Quartiles are statistical measures that divide a sorted dataset into four equal parts, each containing approximately 25% of the observations. In the present study, patients were stratified into quartiles (Q1-Q4) based on the distribution of Lp-PLA2 activity levels across the entire study cohort. The quartile cut-off values were as follows: Q1: ≤450 U/L; Q2: 450–556 U/L; Q3: 556–666 U/L; Q4: >666 U/L. TyG Index was calculated as Ln [Fasting Triglycerides (mg/dL) × Fasting Glucose (mg/dL)/2]. (Line 140 to 141, 175 to 177)

Reference:

Chen Q, Hu P, Hou X, Sun Y, Jiao M, Peng L, Dai Z, Yin X, Liu R, Li Y, Zhu C. Association between triglyceride-glucose related indices and mortality among individuals with non-alcoholic fatty liver disease or metabolic dysfunction-associated steatotic liver disease. Cardiovasc Diabetol. 2024 Jul 4;23(1):232.

  1. The results include "Carotid atherosclerosis (n, %)." What is this parameter? How was it defined? Based on CIMT and carotid plaque? I'm wondering if someone had a high CIMT value and a low carotid plaque (or vice versa), did you also recognize CAS? This needs to be clearly defined.
    Response: Thank you for your kind reminding. We have added the relevant definition in the method section. (Line 158 to 167)

CIMT was measured on the far wall of the common carotid artery. The specific site was approximately 1-2 cm proximal to the carotid bulb. Measurements were performed bilaterally (on both the left and right arteries). The mean value was calculated from all valid measurements averaged across both sides. We explicitly state that measurements were acquired in plaque-free segments, adhering to international consensus guidelines. Intima-media thickening of the carotid artery is defined as a CIMT of ≥1.0 mm. Carotid plaque is characterized as localized thickening of the arterial lumen greater than 0.5 mm or exceeding 50% of the surrounding intima-media thickness. Carotid atherosclerosis was defined as intima-media thickening of the carotid artery or the presence of carotid plaque. (Line 158 to 167)

[1] Touboul PJ, Hennerici MG, Meairs S, Adams H, Amarenco P, Bornstein N, et al. Mannheim carotid intima-media thickness and plaque consensus (2004-2006-2011). An update on behalf of the advisory board of the 3rd, 4th and 5th watching the risk symposia, at the 13th, 15th and 20th European Stroke Conferences, Mannheim, Germany, 2004, Brussels, Belgium, 2006, and Hamburg, Germany, 2011. Cerebrovasc Dis. 2012;34(4):290-296.

[2] Stein JH, Korcarz CE, Hurst RT, Lonn E, Kendall CB, Mohler ER, et al. Use of carotid ultrasound to identify subclinical vascular disease and evaluate cardiovascular disease risk: a consensus statement from the American Society of Echocardiography Carotid Intima-Media Thickness Task Force. Endorsed by the Society for Vascular Medicine. J Am Soc Echocardiogr. 2008 Feb;21(2):93-111.

Round 2

Reviewer 1 Report

Comments and Suggestions for Authors

Dear Authors,

Thank you for your thorough revision and detailed responses to the reviewers’ comments. I appreciate the substantial effort you made to improve the manuscript. Overall, I believe the revisions have adequately addressed the concerns raised in the first review. The manuscript is now clearer, more balanced.

Best regards.

Author Response

Dear Reviewer,

We are most grateful for the time and care the reviewer spent for improving our manuscript.

Please find below the point-by-point responses to your comments. 

Sincerely,

Congxiang Shao and Bihui Zhong

On behalf of the author team

Reviewer 1

Dear Authors,

Thank you for your thorough revision and detailed responses to the reviewers’ comments. I appreciate the substantial effort you made to improve the manuscript. Overall, I believe the revisions have adequately addressed the concerns raised in the first review. The manuscript is now clearer, more balanced.

Best regards.

Reply: We are truly honored to have you as the reviewer of our manuscript and appreciate your valuable contributions to enhancing the quality of our work.